# Calcium Export from Neurons and Multi-Kinase Signaling Cascades Contribute to Ouabain Neuroprotection in Hyperhomocysteinemia

**DOI:** 10.3390/biom10081104

**Published:** 2020-07-24

**Authors:** Maria A. Ivanova, Arina D. Kokorina, Polina D. Timofeeva, Tatiana V. Karelina, Polina A. Abushik, Julia D. Stepanenko, Dmitry A. Sibarov, Sergei M. Antonov

**Affiliations:** Laboratory of Comparative Neurophysiology, Sechenov Institute of Evolutionary Physiology and Biochemistry of the Russian Academy of Sciences, 194223 Saint-Petersburg, Russia; ivanova_ma@iephb.ru (M.A.I.); el-kaa@gmail.com (A.D.K.); severustobsnape@yandex.ru (P.D.T.); karelina_tanja@mail.ru (T.V.K.); polinaabushik@gmail.com (P.A.A.); juli@unixway.org (J.D.S.); dsibarov@gmail.com (D.A.S.)

**Keywords:** homocysteine, glutamate, NMDA receptors, cortical neurons, ouabain, calcium

## Abstract

Pathological homocysteine (HCY) accumulation in the human plasma, known as hyperhomocysteinemia, exacerbates neurodegenerative diseases because, in the brain, this amino acid acts as a persistent *N*-methyl-d-aspartate receptor agonist. We studied the effects of 0.1–1 nM ouabain on intracellular Ca^2+^ signaling, mitochondrial inner membrane voltage (φ_mit_), and cell viability in primary cultures of rat cortical neurons in glutamate and HCY neurotoxic insults. In addition, apoptosis-related protein expression and the involvement of some kinases in ouabain-mediated effects were evaluated. In short insults, HCY was less potent than glutamate as a neurotoxic agent and induced a 20% loss of φ_mit_, whereas glutamate caused a 70% decrease of this value. Subnanomolar ouabain exhibited immediate and postponed neuroprotective effects on neurons. (1) Ouabain rapidly reduced the Ca^2+^ overload of neurons and loss of φ_mit_ evoked by glutamate and HCY that rescued neurons in short insults. (2) In prolonged 24 h excitotoxic insults, ouabain prevented neuronal apoptosis, triggering proteinkinase A and proteinkinase C dependent intracellular neuroprotective cascades for HCY, but not for glutamate. We, therefore, demonstrated here the role of PKC and PKA involving pathways in neuronal survival caused by ouabain in hyperhomocysteinemia, which suggests existence of different appropriate pharmacological treatment for hyperhomocysteinemia and glutamate excitotoxicity.

## 1. Introduction

Homocysteine (2-amino-4-sulfanylbutanoic acid, HCY), a thiol-containing amino acid, represents an intermediate product for methionine and cysteine synthesis, the proper metabolism of which critically depends upon supplies of folic acid and vitamin B_12_. Under physiological conditions, HCY is present in the human plasma in concentrations of 7–14 μM [1]. Both mutations in enzymes of methionine and cysteine synthesis and dietary deficiencies of folic acid and vitamin B_12_ cause an accumulation of HCY in the blood and cerebrospinal fluid. Elevated plasma HCY levels, known as hyperhomocysteinemia, represent a risk factor for stroke [2] and can exacerbate many neuronal disorders, including Parkinson’s [3,4] and Alzheimer’s diseases [5]. In severe hyperhomocysteinemia, the plasma HCY composition can exceed 100 μM [6].

It is well established that excessive HCY concentrations perform neurotoxicity [7,8], causing apoptosis of neurons of different brain regions [9]. The ability of HCY to activate ionotropic glutamate receptors, particularly of *N*-methyl-d-aspartate-type (NMDARs) and metabotropic glutamate receptors of type-5 (mGluR5s), is generally thought to underlay the neurotoxicity [7,10]. The blocking of NMDARs with antagonists prevents HCY neurotoxic effects [10]. When studied at concentrations up to 1 mM, HCY activated NMDAR currents, the amplitudes of which increased with the HCY concentration ([HCY]) increase [7,11]. The effect depended on GluN2 subunit compositions of NMDARs [11]. When studied at physiologically relevant concentrations, it appeared that all diheteromeric NMDARs (containing GluN2A, B, C, or D) were well activated by HCY with EC_50_ varying from about 10 µM (for GluN2A) to 80 µM (for GluN2C) [12,13]. Whereas NMDARs containing GluN2A/C or D subunits did not reveal desensitization during the HCY activation [12,13], currents through GluN1/GluN2B NMDARs briefly declined to about 15% of the peak amplitude because of the desensitization [12]. This observation exhibits why NMDARs containing GluN2A subunits, but not those of GluN2B, mostly contribute to HCY-induced excitotoxicity in cortical neurons [12,13,14,15,16]. Cerebellar neurons expressing NMDARs containing GluN2C or 2D are also sensitive to the excitotoxic action of HCY since these receptors do not reveal desensitization during the HCY activation [13].

To compare neurotoxicity caused by glutamate and HCY, which are both endogenous amino acids, a much broader range of molecular targets for glutamate than for HCY in the brain, including all ionotropic and metabotropic glutamate receptors as well as glutamate transporters, should be taken into account. The Ca^2+^ influx through Ca^2+^-permeable AMPA receptors and synaptic GluN2A- and extrasynaptic GluN2B-containing NMDARs with subsequent ERK MAPK and caspase-3 activation play a role in the glutamate-induced neurotoxicity [17,18,19]. In contrast, HCY activates synaptic GluN2A-containing NMDARs and desensitized extrasynaptic NMDARs, causing excitatory transmission hyperactivity [20,21]. Therefore, the HCY-induced neurotoxicity differs from that of glutamate in many aspects, most likely including intracellular signaling cascades as well.

Previously, we have shown that Ca^2+^ overload of neurons caused by glutamate, NMDA, and kainic acid can be prevented by subnanomolar concentrations of ouabain acting via sodium-potassium ATPase (NKA) [22,23] and by activation of cAMP-dependent signaling [24] that considerably improves neuronal viability. NKA is a critical element of cell ionic balance providing Na^+^ and K^+^ trans-membrane gradients for other electrolyte and organic transporters and ion channels. The value of these processes for cell functioning is highlighted by the observation that NKA enzymatic activity inhibition results in cell death. There is a growing pool of evidence that NKA also acts as a signal transducer [25,26,27,28,29]. Ouabain is a well-studied cardiotonic steroid, which can specifically bind to NKA and inhibit its pump activity. Neurons express α1- and α3-isoforms of NKA. NKAα1 is many-fold less sensitive to ouabain than NKAα3, which is inhibited by ouabain at concentrations above 10 nM [30,31]. Ouabain at 1 nM, however, potentiates ion transport by NKAα3 without any effect on NKAα1 [30]. Thus, sub-saturating concentrations of ouabain (usually 1 nM and below) do not inhibit NKA but activate various metabotropic signaling cascades via Src-kinase [32], IP_3_-receptor [33], Na/Ca-exchanger [22,34], protein kinases phosphatidylinositide 3-kinase, and Akt [35]. At these concentrations, ouabain can regulate Ca^2+^ extrusion from neurons by Na^+^/Ca^2+^-exchangers interacting with NKA in the lipid rafts, which may also influence the functioning of NMDARs [36,37]. Although NKA signaling triggered by low concentrations of cardiotonic steroids results in neuroprotection [22,38], the mechanisms and the particular intracellular cascades involved in these effects are not fully understood. 

In excitotoxic stress, NKA-signaling has both effects on Ca^2+^ transport [22,23] and on apoptotic pathways and neuronal viability [22,38]. Recently it has been demonstrated that calcitonin gene-related peptide (CGRP), which is a mediator of pain in migraine [39], causes neuroprotection of different neurons against HCY by activation of multi-kinase signaling [40]. Here in the model of short (4 h) and long (24 h) insults of glutamate and HCY on cortical neurons in primary culture, we analyze neuronal viability and involvement of kinases in ouabain-induced neuroprotection. We also compare the ouabain effects on glutamate and HCY-elicited intracellular Ca^2+^ responses and changes of mitochondrial inner membrane potential.

## 2. Materials and Methods 

### 2.1. Material and Animals 

Chemicals were from Sigma unless otherwise indicated. All procedures using animals were in accordance with recommendations of the Federation for Laboratory Animal Science Associations and approved by animal care and use committee of Sechenov Institute of Evolutionary Physiology and Biochemistry of the Russian Academy of Sciences (project 16-15-10192, 01.05.2016).

### 2.2. Primary Cortical Culture

The procedure of culture preparation from rat embryos was previously described [41,42]. Briefly, Wistar rats (overall 27 animals provided by the Sechenov Institute’s Animal Facility), 16–17 days pregnant, were sacrificed by CO_2_ inhalation. Fetuses were removed, and their cerebral cortices were isolated, enzymatically dissociated, and used to prepare primary neuronal cultures. Cells were grown in Neurobasal^TM^ culture media supplemented with B-27 (Gibco) on glass coverslips coated with poly-d-lysine for 10–14 days (10–14 DIV) before experiments [41,42].

### 2.3. Quantitation of Cell Viability

To test the viability, cultured cells were incubated for 4 or 24 h in normal medium (control conditions) or in medium containing 100 µM l-glutamate (Glu) or 100 µM l-homocysteine (HCY). Agonists were also combined with 0.1 or 1 nM oubain. These concentrations of ouabain provided neuroprotection against exciotoxicity when studied for dose-dependence [31]. Here both glutamate receptor agonists were co-applied with 30 µM glycine to ensure activation of NMDA receptors. The specific protein kinase A (PKA) inhibitor fragment 14–22 (PKAi), myristoylated trifluoroacetate, was used to inhibit PKA and subsequent cAMP signaling pathways; chelerythrine was applied to inhibit protein kinase C (PKC) and KN93 to inhibit Ca^2+^/calmodulin-dependent protein kinase type II (CaMKII). For simplicity in the further description, we use name “short” and “long” to distinguish between 4 h and 24 h excitotoxicity protocols.

Cell viability was measured by the fluorescent viability assay (FVA) described earlier [41]. Cells were stained with 0.001% acridine orange for 30 s in a basic solution (152 mM NaCl, 2.5 mM KCl, 10 mM glucose, 2 mM CaCl_2_, 1 mM MgCl_2_, 10 mM HEPES, pH adjusted to 7.4 using NaOH). After complete washout of contaminating acridine orange, cells were exposed to 0.004% ethidium bromide for 30 s in the basic solution followed by the dye washout. This staining was performed immediately before each measurement. Fluorescence images were captured using a Leica TCS SP5 MP scanning confocal microscope (Leica Microsystems Inc., Wetzlar, Germany). For two-channel imaging, the fluorescence was excited with a 488 nm laser, and the emitted fluorescence was acquired at 500 to 560 nm (green region of spectrum for acridine orange) and above 600 nm (red region of spectrum for ethidium bromide). Single focal plane images from both channels were merged and analyzed with standard Leica LAS AF software (Leica Microsystems Inc.) and ImageJ software using a custom-written plug-in [22]. On the resulting image, non-colocalized green and red pixels were attributed to live and necrotic neurons, respectively. Yellow–orange pixels with colocalized green and red fluorescence were attributed to the nuclei of apoptotic neurons.

### 2.4. Western Blot Analysis 

The cells were washed twice with cold PBS and lysed in a buffer containing 50 mM Tris-HCl, 150 mM NaCl, 1 mM EDTA, 1 mM EGTA, 10% glycerin, 1% Triton X-100, 1 mM Na_3_VO_4_, 1 mM NaF, 0.5 mM PMSF, and a cocktail of protease inhibitors (1:500, Sigma, USA) for 10 min on ice. After lysis, the cells were scraped off the glasses and centrifuged for 15 min at 15,000 g. One quarter part of the buffer for electrophoretic samples (40 mM Tris, pH 6.8, 10% SDS, 20% 2-mercaptoethanol, and 40% glycerol) was added to the supernatant and incubated for 5 min at 100 °C. Then, 10 µg of each sample were loaded on 10% polyacrylamide gel for electrophoresis and then transferred onto polyvinylidene fluoride membranes in a cold room during the night at 20 V in a small BioRad chamber. The quality of the transfer was checked by marking the gel with Coomassie dye. Non-specific binding sites were blocked with Tris-buffered saline–Tween (0.02 M Tris, 0.137 M NaCl, and 0.1% Tween 20) containing 5% non-fat dried milk and probed overnight for protein of interest with the following primary antibodies: polyclonal rabbit antibodies to protein caspase-3 (dilution 1:1500, Sigma C-8487), monoclonal rabbit antibodies to apoptosis inducing factor (AIF) (dilution 1:1000, abcam ab32516), monoclonal mouse antibodies to BAX (dilution 1:2000, abcam ab5714), polyclonal rabbit antibodies to Bcl-2 (dilution 1:200, abcam ab7973), and monoclonal mouse antibodies to p53 (dilution 1:150, Sigma P-5813). β-Actin protein (monoclonal mouse antibodies, dilution 1:1000, Sigma A3854) served as the control for gel loading. The excess of primary antibodies was washed out and a solution of corresponding horseradish peroxidase-conjugated secondary antibodies was applied for 2 h: anti-rabbit to Cas-3, AIF, and Bcl-2 proteins (1:2000, Dako P0448 dilution) and anti-mouse to BAX and p53 proteins (1:2000, Dako P0447 dilution). Bands were detected by chemiluminescence, visualized on X-ray film, and quantified by densitometry in ImageJ software.

### 2.5. Calcium Imaging 

Cortical neurons in primary culture 10–14 days in vitro (DIV) were loaded with Fluo-8. For this reason neurons were incubated in a basic solution containing 2 µM Fluo-8 acetoxymethyl ester (Fluo-8 AM) at room temperature for 60 min. To remove contaminating dye, cells were perfused with the pure basic solution for 20 min. Then a coverslip with neurons was placed on the stage of a Leica TCS SP5 MP inverted microscope (Leica Microsystems, GmbH, Germany) and permanently perfused with the basic solution at a flow rate of 1.2 mL/min. The setup was equipped by the fast perfusion system, which allowed rapid application of various compounds. HCY (100 µM) or glutamate (100 µM) were added together with 30 µM glycine. Fluorescence was excited with 488 nm laser and detected at a 510–560 nm range with ~2 s sampling interval (frame 512 × 512 px).

### 2.6. Mitochondrial Membrane Potential Imaging 

Cortical neurons were incubated within the basic solution containing 5 µM rhodamine-123 for 30 min. The dye fluorescence was detected using a Leica TCS SP5 MP inverted microscope (Leica Microsystems Inc.). The wavelengths of 488 nm and 510–530 nm were used for excitation and emission, respectively, to monitor mitochondrial inner membrane potential (φ_mit_). The sampling interval was set to ~1 s (frame 512 × 512 px). In experiments, HCY (100 µM) or glutamate (100 µM) were always co-applied with 30 µM glycine. To test the functional state of mitochondria, the oxidative phosphorylation inhibitor, 4 µM CCCP (m-chlorophenyl hydrazone, Sigma, St. Louis, MO, USA) was added at the end of experiment [43].

### 2.7. Analysis and Statistics

In the text, *n* represents the number of experiments, where each of experiments was performed on a single coverslip with cultured neurons. Groups comparison was done using ANOVA with Tukey’s post-*hoc* test and Student’s *t*-test. Each group contained coverslips from at least 3 different cultures. The results are expressed as mean values ± SEM unless otherwise stated. The level of statistical significance was set to *p* < 0.05. Data marked by *, **, or *** are significantly different with *p* < 0.05, *p* < 0.01, and *p* < 0.001, respectively.

## 3. Results

### 3.1. Ouabain Enhances the Viability of Neurons in Short Excitotoxic Stress

The neuroprotective effect of ouabain was assessed in the model of excitotoxic stress elicited by 4 h incubation of neurons in the bathing solution containing 100 μM glutamate or 100 μM HCY (Figure 1). The fraction of live cells (Figure 1A) was significantly decreased (*p* = 0.0001, *n* = 8) due to apoptosis (*p* = 0.0001, *n* = 8) and necrosis (*p* = 0.002, *n* = 8) after 4 h incubation in the 100 μM glutamate-containing bathing solution (Figure 1B). The neuronal viability was found to be unchanged from control values if glutamate was co-applied with 0.1 or 1 nM ouabain (Figure 1C,D). A similar neurotoxic effect was elicited by 4 h incubation in the bathing solution with 100 μM HCY, while the addition of 0.1 or 1 nM ouabain prevented necrosis and apoptosis of neurons (Figure 1E). From these experiments, we may conclude that 4 h treatments with equimolar concentrations of either glutamate or HCY resulted in apoptosis and necrosis (Table 1). Subnanomolar concentrations of ouabain, however, prevented the neurotoxic effects of these compounds. Because apoptosis was more pronounced than necrosis in the effects of both amino acids, we further analyzed the influence of ouabain on the expression of pro- and anti-apoptotic proteins as well as on calcium and energy balance in neurons.

### 3.2. Ouabain Prevents Pro-Apoptotic Proteins Expression in Short Excitotoxic Insults

In terms of ouabain inhibited neuronal apoptosis, we studied whether the expression of proteins involved in apoptosis pathways and anti-apoptotic proteins was affected by ouabain during 4 h neurotoxic action of glutamate. We studied expression of proteins such as caspase-3 (Cas-3), p53, and BAX, which are involved in the caspase-dependent apoptotic pathway. The expression of apoptosis-inducing factor (AIF) was also evaluated together with Bcl-2, which antagonizes AIF release from mitochondria.

Western blotting confirmed that AIF, Cas-3, p53, and BAX expression were elevated, and Bcl-2 expression was suppressed after 4 h incubation in the bathing solution containing 100 μM glutamate (Figure 2). The glutamate-containing bathing solution combined with 1 nM ouabain prevented an increase of pro-apoptotic protein expression, and the Bcl-2 expression remained at the control level. From these experiments, it became clear that ouabain at 1 nM could antagonize the development of apoptotic signaling pathways so that the pattern of protein expression in the presence of glutamate and ouabain was found to be similar to the control conditions (Figure 2).

### 3.3. Ouabain Suppresses Intracellular Ca^2+^ Overload and Mitochondrial Dysfunction of Neurons

It is generally accepted that the Bcl-2 level is positively correlated with normal mitochondria functioning and energy metabolism. A decrease in Bcl-2 levels is widely used as a marker of apoptosis [44]. Considering that ouabain restores Bcl-2 expression in excitotoxic insults, in further experiments we used fluorescent imaging to monitor free intracellular Ca^2+^ concentration ([Ca^2+^]) and changes of the internal membrane potential of mitochondria in neurons during applications of glutamate or HCY in the absence or in the presence of 1 nM ouabain.

The intracellular Ca^2+^ responses elicited by 100 μM glutamate application (Figure 3A,B) demonstrated the permanent Ca^2+^ elevation as long as glutamate was present. Some neurons generated Ca^2+^ transients in response to glutamate (Figure 3C), which contributed to intracellular [Ca^2+^] elevation. Nevertheless, application of 1 nM ouabain reduced the glutamate-evoked increase of intracellular [Ca^2+^] (Figure 3B) and suppressed Ca^2+^ transients (Figure 3C) that prevented cytosolic Ca^2+^ accumulation (Figure 3D). Therefore, in the presence of ouabain, the averaged glutamate elicited [Ca^2+^] elevation did not exceed control values (*n* = 5, Figure 3B,C), being strongly reduced as compared to those obtained in the absence of ouabain (Figure 3D, *p* = 0.0006, *n* = 5).

The Ca^2+^ entry in neurons in response to glutamate exposure was accompanied by the significant loss of internal membrane voltage of mitochondria (Figure 3E), which reflects the energy cost for excessive intracellular Ca^2+^ removal from the cytoplasm by intracellular Ca^2+^ stores including mitochondria and ion transport mechanisms. The voltage loss (∆φ_mit_) reached 0.7 of the maximal level (φ_mit_) estimated by CCCP, which, as is known, causes a mitochondrial respiratory chain disruption resulting in the ATP synthesis collapse and the energy depletion. In the presence of ouabain, glutamate did not evoke any significant loss of inner membrane voltage of mitochondria (Figure 3F).

Therefore short insults of glutamate caused the intracellular Ca^2+^ accumulation in neurons (Figure 3D), which may have reached over 500% of control (*p* = 0.0006, *n* = 7, Figure 3G), and mitochondrial dysfunction (Figure 3E), which was characterized by ∆φ_mit_ = 0.7 (Figure 3G). In the presence of 1 nM ouabain, these glutamate effects were not observed (Figure 3G).

Similar results were obtained in experiments with 100 μM HCY, which, however, differed from the glutamate effects quantitatively. The permanent three-fold elevation of intracellular [Ca^2+^] evoked by HCY (*p* = 0.0004, *n* = 4, Figure 4A) was significantly suppressed by ouabain (*p* = 0.0002, *n* = 4, Figure 4B), which abolished cytosolic Ca^2+^ accumulation (Figure 4C). The ∆φ_mit_ value for HCY was found to be 0.2 (*p* = 0.0002, *n* = 5, Figure 4D), which was much smaller than those for Glu (*p* = 0.0006, *n* = 7, Figure 3G) and was not observed in the presence of 1 nM ouabain (Figure 4E). Therefore HCY caused smaller Ca^2+^ responses and produced a less burdening effect on the cell energy balance than Glu (Figure 4F). Ouabain effectively prevented both the HCY evoked Ca^2+^ accumulation and the drop of mitochondrial inner membrane potential.

Thus, at 100 μM, both studied amino acids caused the intracellular [Ca^2+^] elevation and accompanying loss of the mitochondrial inner membrane voltage. While the effects of HCY were less pronounced than for glutamate, ouabain was able to prevent the effects of both amino acids.

### 3.4. Ouabain Enhances the Viability of Neurons in Long Excitotoxic Stress Triggering Multi-Kinase Signaling Cascades

Long-term 24 h incubation of neurons in the presence of 100 μM Glu (Figure 5A) as well as of 100 μM HCY (Figure 5B) resulted in an increase of neuronal apoptosis (*n* = 8, *p* = 0.03 for Glu and *n* = 8, *p* = 0.0008 for HCY). Combined applications of 0.1 nM–1 nM ouabain with Glu or HCY completely prevented apoptosis induced by both NMDAR agonists (Figure 5C,D). Quantitative comparison of the data obtained for Glu and HCY and in the presence of 0.1 nM or 1 nM ouabain are shown in Figure 5E and in Table 2.

To evaluate the intracellular anti-apoptotic cascades triggered by ouabain, we further studied the involvement of protein kinases in these processes using their selective inhibitors. The effects of PKA inhibition by PKAi (0.6 μM), PKC inhibition by Chel (1 μM), or CaMKII inhibition by KN93 (3 μM) were studied in short- and long-term protocols of the excitotoxic stress experiments. The short-term 4 h incubation of neurons with agonists + 1 nM ouabain and combined with kinase inhibitors did not substantially change neuronal survival and apoptosis (Figure 6A). From this observation, we may conclude that short-term anti-apoptotic effects of ouabain do not depend on studied kinases and probably involve some fast-acting “emergency” mechanisms. In 24 h treatment of neurons with agonists, some differences between Glu and HCY effects were found. For example, the inhibition of all kinases under study had no considerable effects on ouabain-caused neuroprotection against glutamate neurotoxicity. In contrast, the inhibition of either PKA or PKC prevented anti-apoptotic effects of ouabain during 24 h treatment with HCY (Figure 6B). KN93 did not prevent neuroprotection of ouabain in experiments with HCY, suggesting that CaMKII is not involved in this process. From these observations, we may conclude that in the condition of hyperhomocysteinemia, both PKA and PKC are required for anti-apoptotic signaling pathways triggered by ouabain.

## 4. Discussion

It is widely accepted that Glu- [16,18,19,45] and HCY-induced [7,8] neurodegeneration is caused by the permanent plasma membrane depolarization and cytosolic Ca^2+^ overload of neurons, which is the condition known as excitotoxic stress (for a review see [45]). Under excitotoxic stress, the maintenance of ionic gradients on the plasma membrane by NKA consumes ATP and burdens mitochondria, which, as a consequence, is followed by the loss of mitochondrial inner membrane voltage. This particular situation could be illustrated by the disruption of the mitochondrial respiratory chain with CCCP and subsequent total loss of inner membrane voltage. Furthermore, water entry via ionotropic receptor channels to balance an accumulation of ions inside of neurons and a low Na^+^ membrane gradient can result in neuroinflammation with fast necrosis of neurons. In addition, long-term elevation of cytosolic Ca^2+^ and mitochondrial dysfunction can trigger intracellular signaling cascades of apoptosis. Overall, our data concerning the effects of Glu and HCY on neurons are consistent with this concept of excitotoxicity. We therefore further consider the mechanisms of neuroprotection by a subnanomolar concentration of ouabain.

### 4.1. Necrosis at Short-Term Excitotoxic Stress

The short-term (4 h) action of both Glu and HCY resulted in a considerable increase of necrotic neurons. This effect was abolished by the combined application of agonists with 0.1 nM or 1 nM ouabain. Presumably, in a portion of neurons, the plasma membrane depolarization and Ca^2+^ overload evoked by Glu or HCY engage energy-consuming ionic homeostasis, which leads to a rapid depletion of ATP resources required for the maintenance of ion balance and coupled water transport. In experiments, these processes could be monitored by the loss of mitochondrial inner membrane potential. The above explanation is consistent with the earlier observation that NMDAR-induced osmotic imbalance causes cell swelling [46,47] and rapid necrosis of neurons [46].

In short-term experiments, HCY, as an excitotoxic agent, was less potent than Glu. This conclusion is supported by the observations that (1) the Ca^2+^ overload of neurons induced by HCY was significantly less pronounced than as for Glu (***, *p* = 0.0001, *n* = 7, Student’s *t*-test), and (2) HCY caused a significantly lesser drop of mitochondrial inner membrane voltage, which was found to be about 0.2 (∆φ_mit_) of the total drop (φ_mit_) caused by CCCP, than the ∆φ_mit_ value of about 0.7 found for Glu (***, *p* = 0.0001, *n* = 5, Student’s *t*-test). As is known, these agonists differ with respect to the activation of Glu receptors. While Glu activates all Glu receptors and does not cause desensitization of NMDARs, HCY activates only NMDARs and desensitizes those containing GluN2B subunits [12]. The latter ones are widely expressed in extrasynaptic regions of the plasma membrane and are thought to provide a major contribution to neurodegeneration [19]. Since rat cortical neurons express GluN2A and GluN2B (for a review see [48]), the HCY effects are likely determined preferentially by the activation of GluN2A-containing NMDARs [12,16,20]. In contrast, Glu also activates GluN2B-containing NMDARs, producing more Ca^2+^ to be accumulated in the cytoplasm and a pronounced drop of the mitochondrial inner membrane potential, which is usually associated with mitochondrial swelling and neuronal cell death [49].

Subnanomolar concentrations of ouabain applied with either Glu or HCY considerably reduced the intracellular Ca^2+^ accumulation and related drop of the mitochondrial inner membrane potential, which rescued neurons from ATP deficit and reduced necrosis of neurons (Figure 7A). This emergency rescue of neurons is most likely determined by an acceleration of Ca^2+^ export from neurons by NCXs, which, as demonstrated previously, are functionally regulated by ouabain through interaction with NKAs [22]. Besides, ouabain considerably decreased Ca^2+^ transients generated in some neurons on agonist application, which coincides well with the previous data that show that ouabain lowers the frequency of spontaneous excitatory postsynaptic currents in cortical neurons [22]. In addition, amplitudes of Ca^2+^ transients are controlled by the local interplay of ouabain sensitive α3NKA, NCX, and pre-membrane endoplasmic reticulum [22,50]. It seems unlikely that this mechanism of intracellular Ca^2+^ regulation is specific for NMDARs because, as shown earlier, an increase of intracellular [Ca^2+^] elicited by kainite activation of AMPA/kainite receptors is also well prevented by 1 nM ouabain [23].

### 4.2. Apoptosis at Short-Term Excitotoxic Stress

Short excitotoxic insults induced by Glu in our experiments caused the increase of p53, AIF, and Bax, which are pro-apoptotic proteins, and expression and Cas-3 activation in neurons, which was accompanied by loss of Bcl-2. Presumably, Glu elicited elevation of intracellular [Ca^2+^], and loss of mitochondrial inner membrane voltage led to the opening of the Bcl-2 controlled permeability transition pore of mitochondria and releases of AIF, cytochrome-C, and other pro-apoptotic factors that initiate apoptosis [51]. Consistence with previous studies, we suggest that 4 h Glu insults cause an elevation of p53 expression resulting in a subsequent Bax activation [52], which may enhance NMDA-elicited Ca^2+^ transients and contribute to deregulation of φ_mit_ [53]. Previously it was demonstrated that ouabain at nanomolar concentrations causes a reduction of p53 expression by activation of Src/mitogen-activated protein kinase (MAPK) signaling pathways upon its binding to the NKA [54]. Therefore, ouabain-induced signaling may prevent the up-regulation and mitochondrial recruitment of Bax [55,56], which opposes Ca^2+^-induced mitochondrial dysfunction and apoptosis (Figure 7B).

Cas-3 is protease-activated by both extrinsic and intrinsic (mitochondrial) apoptotic pathways [51] and acts at late irreversible stages of apoptosis. Prevention of Cas-3 activation by ouabain during 4 h excitotoxic insults could be caused by the inactivation of up-stream pro-apoptotic signaling cascades.

Regardless of the observations that Glu effects on Ca^2+^ accumulation and the loss of mitochondrial inner membrane voltage are more pronounced than the effects of HCY, both Glu and HCY caused a similar increase of neuronal apoptosis after 4 h treatment. Ouabain also produced similar protection against apoptosis and necrosis in the case of 4 h incubation with both HCY and Glu. By the use of specific inhibitors, we show that PKC, PKA, or CaMKII are not involved in ouabain neuroprotection at 4 h excitotoxic stress. Most likely, anti-apoptotic mechanisms such as CREB phosphorylation by PKA or PKC [57] do not contribute to the ouabain effects in the short excitotoxic insults.

### 4.3. Apoptosis at Long-Term Excitotoxic Stress

The long-term 24 h treatment of neurons with Glu or HCY substantially increased the fraction of apoptotic but not necrotic neurons. Ouabain at 1 nM effectively prevented neuronal apoptosis against 24 h treatments by both Glu and HCY. Inhibition of CaMKII had no effect on ouabain mediated anti-apoptotic action in contrast to the effects of CGRP and forskolin [40,58].

The inhibition of PKA or PKC blocked ouabain-induced neuroprotection against HCY but not against Glu neurotoxicity. This observation can be related to the NMDAR subtype selectivity of HCY because GluN2A, but not GluN2B NMDAR subunits, mostly contribute to HCY toxicity in cortical neurons [12,14,15,16] and reflect extensive recruitment of different glutamate ionotropic and metabotropic receptors and transporters in glutamate neurotoxicity.

This indirectly supports the idea that HCY activates unique pro-apoptotic mechanisms that differ from those induced by Glu. For example, during long-term action, HCY, but not Glu, induces GluN2A-dependent sustained activation of ERK2 MAPK [8,59], internalization of the Ca^2+^-impermeable GluA2-subunit of AMPA receptors, and increase of intracellular [Ca^2+^] [60], which both cause permanent activation of p38 MAPK [14], downstream phosphorylation of caspase-9, and apoptosis (for a review see [61]). Conversely, GluN2B-containing extrasynaptic NMDARs over-activated by Glu induce only transient [59] ERK MAPK activation [17,18,19], while HCY causes sustained p38 MAPK activity [14]. As a result, p38 MAPK inhibition can protect neurons from short-term but not long-term activation of NMDARs [62]. In addition, in the case of HCY, the toxicity caused by both 4 h [14] and 24 h [63] treatments can be prevented by p38 MAPK block. Taken together, these observations favor the assumption that NKA signaling could modulate p38 MAPK via some PKA- or PKC-dependent pathways.

### 4.4. The Role of PKC and PKA in Ouabain Effects against Hyperhomocysteinemia

Previously, we demonstrated that neuronal apoptosis induced by 24 h treatment with HCY could be prevented by CGRP or forskolin, which both activated anti-apoptotic cAMP-dependent pathways. This type of neuroprotection depended mostly on PKA and CaMKII activity [40]. Ouabain-induced neuroprotection against 24 h HCY treatment of cortical neurons depended on PKA and PKC, but not CaMKII, the activity of which is mostly linked to the postsynaptic area (for a review see [64]). Therefore, the long-term neuroprotection caused by ouabain must involve the apoptotic mechanisms, which are not yet activated at 4 h but already contribute to neurotoxicity at 24 h excitotoxic stress. The ouabain effects against long-term action of HCY can involve PKA- and PKC-dependent effects on the pro-apoptotic cascade up-regulated by HCY but not by Glu. Probably, the HCY-specific pro-apoptotic cascade [8,59] includes caspase-9 and p38 MAPK, which both are susceptible to inactivation by phosphorylation with PKC [65,66] (Figure 7C), while caspase-9 can be inactivated by PKA or PKC to interrupt apoptosis [67] (Figure 7C,D). Subnanomolar ouabain activates PKC [68], which provides a possible explanation for observed ouabain-induced neuroprotection in hyperhomocysteinemia.

## 5. Conclusions

Effects of short-term treatments of cortical neurons with Glu and HCY differ in many aspects of neurotoxic action, including the intensity of cytosolic Ca^2+^ accumulation and the loss of mitochondrial inner membrane voltages. However, these compounds turned out to be similar with respect to their efficacy to cause necrosis and apoptosis of neurons found in 4 h excitotoxic insults. Under these conditions, the neuroprotection results from the NKA-mediated acceleration of Ca^2+^ exported from neurons that avoids the mitochondrial dysfunction and prevents the development of mitochondrial and caspase-dependent pro-apoptotic cascades. At this particular stage of HCY neurotoxic insult, PKA, PKA, and CaMKII are not necessarily involved in the neuroprotection caused by ouabain, or 4 h insult duration is not enough for these kinases to be activated. Most likely, the improvement of neuronal ionic balance produced by NKA/NCX-mediated signaling is sufficient to prevent both the development of necrosis and apoptosis.

Ouabain effects during long-term 24 h treatment with Glu still do not involve kinase-dependent signaling, while PKC and PKA activity is required for ouabain effects to prevent apoptosis caused by HCY. Unlike Glu, HCY is able to induce the specific GluN2A subunit-dependent sustained activation of the ERK/p38/Cas-9/Cas-3 pathway. This pro-apoptotic pathway is susceptible to the inhibitory phosphorylation by PKC and PKA but does not depend on CaMKII.

## Figures and Tables

**Figure 1 biomolecules-10-01104-f001:**
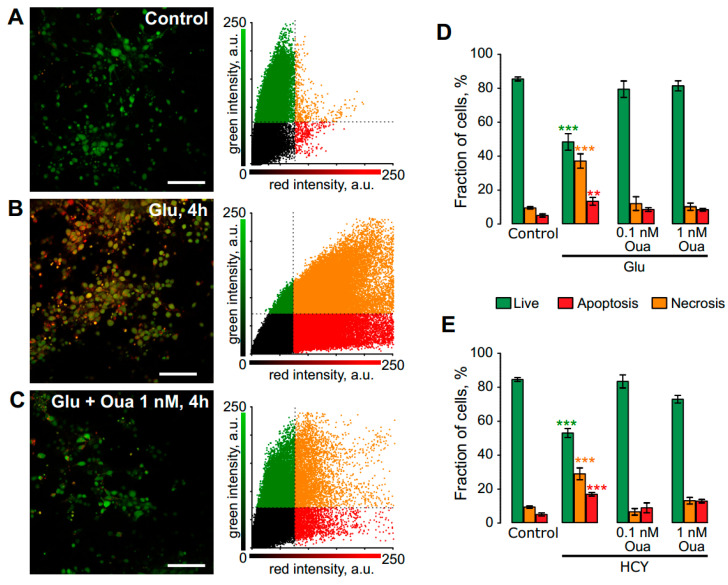
Neuroprotective effect of ouabain (Oua) against neurotoxicity induced by 4 h excitotoxic insults in rat cortical neurons. (**A**–**C**) Confocal images represent an overlay of those recorded in green and red spectral regions of rat cortical neurons after 4 h treatment with the bathing solution (control, panel **A**), and 100 μM glutamate + 30 μM glycine (Glu, panel **B**) and Glu with 1 nM ouabain (Oua, **C**) obtained with the fluorescent viability assay. Scale bar, 100 µm. The correlation plots shown on the right of corresponding images allow estimation of the cell viability (percentages of live, necrotic and apoptotic neurons). (**D**) Histogram quantitatively compares the data obtained in control (*n* = 8) and in the presence Glu (*n* = 7) along and with 0.1 nM (*n* = 8) or 1 nM (*n* = 9) ouabain (Oua). Experimental conditions are indicated below the plots. Green columns represent percentages of live, red—of necrotic and orange—of apoptotic neurons. Mean values ± SEM are plotted. Experimental conditions are indicated below the plots. ** (*p* = 0.002), *** (*p* = 0.0001)—data are significantly different from control by one-way ANOVA with Tukey’s post-*hoc* test. (**E**) Histogram quantitatively compares the data obtained in control (*n* = 8) and in the presence of 100 μM homocysteine + 30 μM glycine (HCY, *n* = 6) and 0.1 (*n* = 7) nM or 1 nM (*n* = 8) ouabain (Oua). Mean values ± SEM are plotted. Experimental conditions are indicated below the plots. Green columns represent percentages of live, red of necrotic, and orange of apoptotic neurons. *** (*p* = 0.0002)—data are significantly different from control by one-way ANOVA with Tukey’s post-*hoc* test, *n* = 7.

**Figure 2 biomolecules-10-01104-f002:**
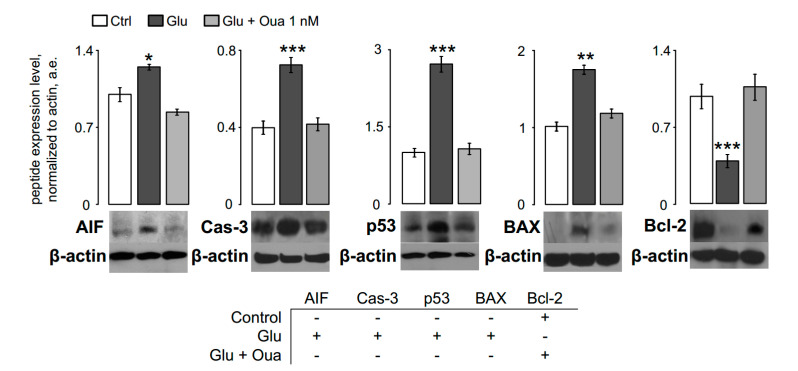
Ouabain prevents the expression of pro-apoptotic proteins induced by 4 h glutamate (Glu) treatment of cortical neurons. Glutamate inhibits expression of Bcl-2 and enhances expression of AIF, Cas-3, p53, and BAX. Co-application of Glu with 1 nM ouabain (Oua) prevents pro-apoptotic protein expression so that the profile of protein expression remains similar to that of control values. Representative images of Western blots showing the protein expression in control, after 4 h treatment with 100 μM glutamate + 30 μM glycine and with 100 μM glutamate + 30 μM glycine + 1 nM ouabain. For quantitative analysis, the blots were scanned, and the intensities of bands after normalizing to β-actin were plotted as means ± SEM (*n* = 6 for each bar). One-way ANOVA with Tukey’s post-*hoc* test was utilized to reveal the significance of difference from control (* *p* = 0.03; ** *p* = 0.005; *** *p* = 0.0002).

**Figure 3 biomolecules-10-01104-f003:**
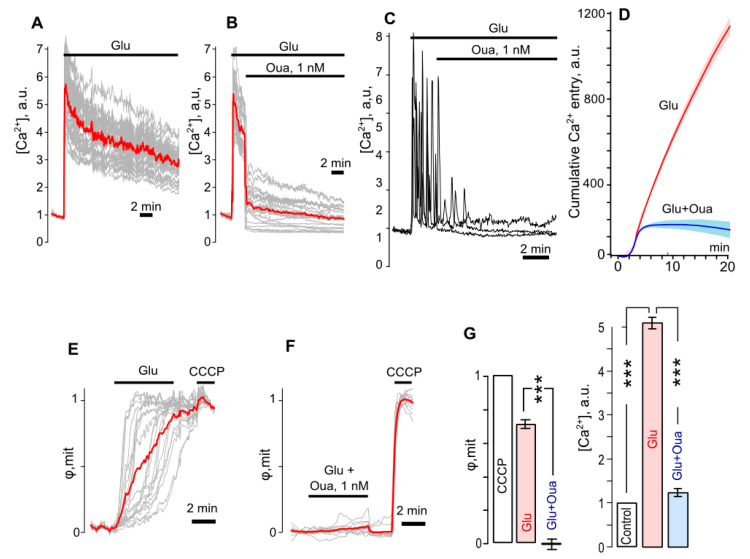
Ouabain prevents the intracellular Ca^2+^ overload and the loss of mitochondrial inner membrane potential of cortical neurons caused by glutamate. (**A**) Fluorescent Ca^2+^ responses of neurons loaded with Fluo-8 evoked by an application of 100 μM glutamate + 30 μM glycine (Glu) obtained from the same experiment and normalized to the fluorescence intensity recorded without Glu. Gray lines—responses of single neurons. Red line—an average response. (**B**) Ca^2+^ responses when 1 nM Oua was added on the top of Glu responses. (**C**) Example of Glu responses revealing generation of Ca^2+^ transients by neurons. (**D**) Average cumulative curves for Ca^2+^ overload evoked Glu (red line) and with an addition of Oua (blue line), which represent an integral of fluorescent Ca^2+^ responses shown in panels (**A**,**B**). Mean value ± SEM for each point is plotted (*n* = 7). (**E**) Fluorescent responses of neurons loaded with rhodamine-123 evoked by an application of 100 μM glutamate + 30 μM glycine (Glu) that reflect the loss of mitochondrial inner membrane voltage (φ_mit_) obtained from the same experiment and normalized to the fluorescence intensity recorded in the presence of 4 µM CCCP. Gray lines—responses of single neurons. Red line—an average response. (**F**) Changes of φ_mit_ when 1 nM Oua was added to Glu. (**G**) Histograms compare average values of φ_mit_ obtained with Glu and combined application of Glu and Oua in relation to full loss of φ_mit_ in the presence of CCCP (on the left, *n* = 7) and Ca^2+^ response amplitudes obtained under control and with Glu and combined application of Glu and Oua (on the right, *n* = 7). ***—Data are significantly different (*p* = 0.0006, ANOVA with Tukey’s post-*hoc* test).

**Figure 4 biomolecules-10-01104-f004:**
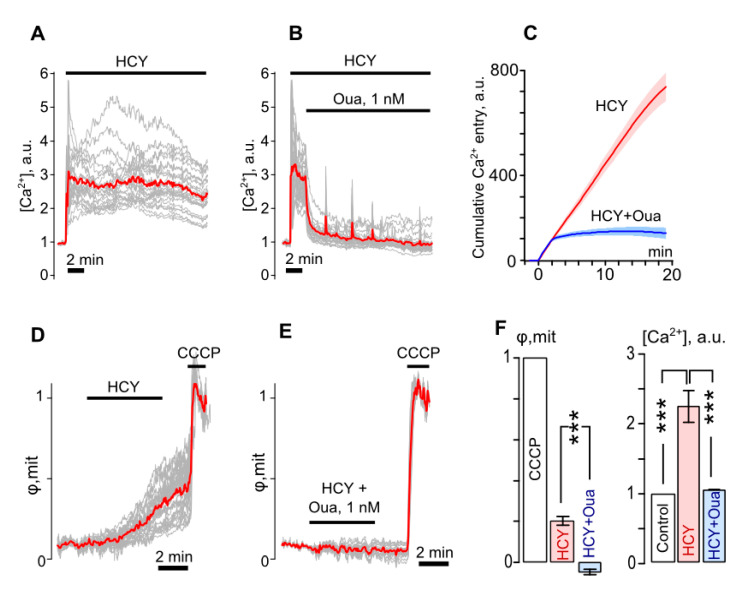
Ouabain prevents the intracellular Ca^2+^ overload and the loss of mitochondrial inner membrane potential of cortical neurons caused by homocystein. (**A**) Fluorescent Ca^2+^ responses of neurons loaded with Fluo-8 evoked by an application of 100 μM homocystein + 30 μM glycine (HCY) obtained from the same experiment and normalized to the fluorescence intensity recorded without HCY. Gray lines—responses of single neurons. Red line—an average response. (**B**) Ca^2+^ responses when 1 nM Oua was added on top of HCY responses. (**C**) Average cumulative curves for Ca^2+^ overload evoked HCY (red line) and with an addition of Oua (blue line), which represent an integral of fluorescent Ca^2+^ responses shown in panels (**A**,**B**). Mean value ± SEM for each point is plotted (*n* = 4). (**D**) Fluorescent responses of neurons loaded with rhodamine-123 evoked by an application of 100 μM homocystein + 30 μM glycine (HCY), which reflect the loss of mitochondrial inner membrane voltage (φ_mit_) obtained from the same experiment and normalized to the fluorescence intensity recorded in the presence of 4 µM CCCP. Gray lines—responses of single neurons. Red line—an average response. (**E**) Changes of φ_mit_ when 1 nM Oua was added to HCY. (**F**) Histograms compare average values of φ_mit_ obtained with HCY and combined application of HCY and Oua in relation to full loss of φ_mit_ in the presence of CCCP (on the left, *n* = 5) and Ca^2+^ response amplitudes ([Ca^2+^]) obtained under control and with Glu and combined application of HCY and Oua (on the right, *n* = 5). ***—Data are significantly different (*p* = 0.0004, ANOVA with Tukey’s post-*hoc* test).

**Figure 5 biomolecules-10-01104-f005:**
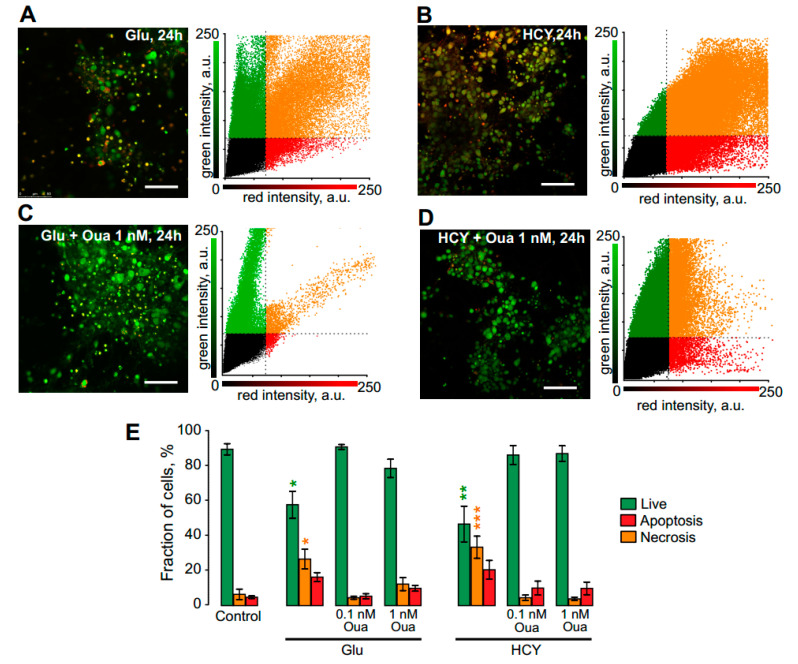
Ouabain prevents the neurotoxicity induced by 24 h excitotoxic insult in rat cortical neurons. (**A**) Confocal image after 24 h treatment with 100 μM Glu + 30 μM glycine (Glu) and (**B**) with 100 μM HCY + 30 μM glycine (HCY); (**C**,**D**) addition of 1 nM ouabain. Scale bar, 100 μm. Data obtained from the control and that after treated with 100 μM Glu or 100 μM HCY and either 0.1 nM or 1 nM Oua. The correlation plots shown on the right of corresponding images allow estimation of the cell viability (percentages of live, necrotic, and apoptotic neurons). (**E**) Data obtained from the control and that after treatment with 100 μM Glu or 100 μM HCY and either 0.1 nM or 1 nM Oua. Green columns: percentages of live. Red: necrotic. Orange: apoptotic neurons. Data are expressed as mean ± SEM. * *p* = 0.03; ** *p* = 0.002; *** *p* = 0.0008, respectively, are significantly different from the control, one-way ANOVA with Tukey’s post-*hoc* test).

**Figure 6 biomolecules-10-01104-f006:**
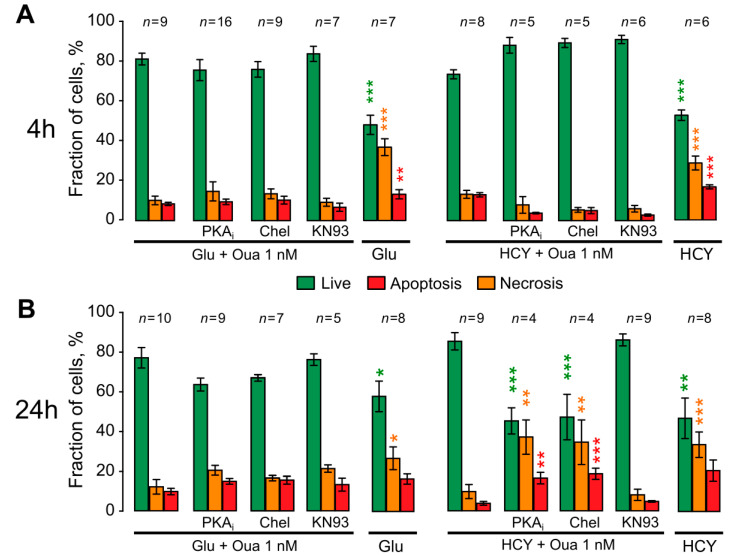
Effects of inhibitors of different protein kinases in ouabain-induced neuroprotection. The histograms quantitatively compare the data obtained in the presence of 100 μM glutamate (+30 μM glycine, Glu) or homocysteine (+30 μM glycine, HCY) with 1 nM ouabain (Oua), and in combination with 0.6 μM PKA inhibitor (PKA_i_), 1 μM PKC inhibitor chelerythrine (Chel), or 3 μM CaMKII inhibitor (KN93) or during excitotoxic insults without Oua. Mean values ± SEM are plotted. Experimental conditions are indicated below the plot. Green columns represent percentages of live, red of necrotic, and orange of apoptotic neurons. Data were obtained in 4 h (**A**) and in 24 h (**B**) excitotoxic insults. *, **, ***—Data are significantly different from the values obtained in HCY with Oua (* *p* = 0.03; ** *p* = 0.009; *** *p* = 0.0001; ANOVA with Tukey’s post-*hoc* test). The number of experiments (*n*) for each group is indicated above the columns.

**Figure 7 biomolecules-10-01104-f007:**
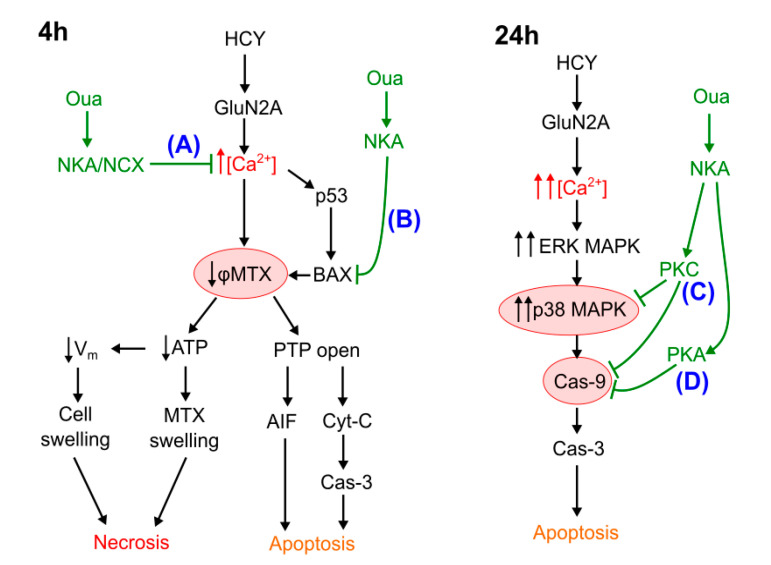
Schematics of data interpretation of antagonism between hyperhomocysteinemia-induced neuronal death and ouabain triggered NKA signaling. The 4 h ouabain effects are kinase-independent and include (**A**) acceleration of Ca^2+^ removal from the cell by NCX, and (**B**) prevention of Bax translocation to mitochondria. Both (**A**) and (**B**) prevent severe ionic imbalance in neurons and augment downstream necrotic and apoptotic neuronal death. The 24 h ouabain effects additionally involve PKC- (**C**) and PKA-dependent (**D**) inhibition of HCY-specific ERK/p38 MAPK apoptotic pathways. [Ca^2+^]—free intracellular calcium concentration; AIF—apoptosis inducing factor; ATP—adenosine-3-phosphate; CaMKII—Ca^2+^/calmodulin-dependent protein kinase II; Cas-3 and Cas-9—caspases 3 and 9, respectively; Cyt-C—cytochrome C; GluN2A—subunit of *N*-methyl-d-aspartate receptors; HCY—homocysteine; MAPK—mitogen-activated protein kinase; MTX—mitochondria; NCX—sodium–calcium exchanger; NKA—Na/K-ATPase; Oua—ouabain; PKA—protein kinase A; PKC—protein kinase C; PTP—mitochondrial permeability transition pore; φMTX—mitochondrial inner membrane potential, V_m_—cell membrane potential.

**Table 1 biomolecules-10-01104-t001:** The neuroprotective effect of ouabain against neurotoxicity induced by 4 h glutamate (Glu) or homocysteine (HCY) treatment of rat cortical neurons.

	Ouabain		Live, %	Apoptotic, %	Necrotic, %
Control		*n* = 8	85 ± 1.2	9 ± 0.7	5 ± 0.9
	+0.1 nM	*n* = 5	85 ± 2.3	9 ± 0.5	6 ± 1.9
	+1 nM	*n* = 7	84 ± 2.6	10 ± 1.1	2 ± 1.7
Glu		*n* = 7	48 ± 4.9 ***	37 ± 4.2 ***	14 ± 2.3 **
	+0.1 nM	*n* = 8	79 ± 4.9	12 ± 4.1	9 ± 1.1
	+1 nM	*n* = 9	81 ± 2.9	10 ± 2.2	8 ± 0.8
HCY		*n* = 6	54 ± 2.7 ***	29 ± 3.5 ***	17 ± 1 ***
	+0.1 nM	*n* = 7	84 ± 3.9	7 ± 1.9	9 ± 2.9
	+1 nM	*n* = 8	74 ± 2.3	13 ± 2	13 ± 1.1

**, ***—data are significantly different from control by one-way ANOVA with Tukey’s post-*hoc* test.

**Table 2 biomolecules-10-01104-t002:** The neuroprotective effect of ouabain against neurotoxicity induced by 24 h glutamate (Glu) or homocysteine (HCY) treatment of rat cortical neurons.

	Ouabain		Live, %	Apoptotic, %	Necrotic, %
Control		*n* = 4	89 ± 3.2	6 ± 3.0	5 ± 0.7
	+1 nM	*n* = 10	78 ± 5.2	12 ± 3.7	10 ± 1.6
Glu		*n* = 6	57 ± 7.6 *	27 ± 5.6 *	16 ± 2.6
	+0.1 nM	*n* = 8	90 ± 1.4	5 ± 0.9	5 ± 1.5
	+1 nM	*n* = 10	78 ± 5.0	11 ± 3.1	11 ± 4.0
HCY		*n* = 5	46 ± 10.1 **	33 ± 6.3 ***	21 ± 5.3
	+0.1 nM	*n* = 9	86 ± 5.4	4 ± 1.6	10 ± 1.6
	+1 nM	*n* = 9	86 ± 4.4	4 ± 0.9	10 ± 3.5

*, **, ***—Data are significantly different from control by one-way ANOVA with Tukey’s post-*hoc* test.

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
