# Peer review of "Calcium Export from Neurons and Multi-Kinase Signaling Cascades Contribute to Ouabain Neuroprotection in Hyperhomocysteinemia"

_biomolecules, 2020, doi:10.3390/biom10081104_

Round 1

Reviewer 1 Report

Recommendation letter to editor Biomolecules

Title: Calcium export from neurons and multi-kinase signaling cascades contribute to ouabain neuroprotection in hyperhomocysteinemia

Authors: Maria A. Ivanova, Arina D. Kokorina, Polina D. Timofeeva, Tatiana V. Karelina, Polina A. Abushik, Julia D. Stepanenko, Dmitry A. Sibarov and Sergei M. Antonov

General Comments:

  1. The written English has to be improved. Many points are not up to the English command. To name few and not limited these examples: Lines 12-15.
  2. Please indicated in the “Abstract” certain sections (although hidden) involving the background, aim, and methods, results & discussion. and conclusions.

Specific Comments:

Line 93-96/Section 2 “Materials and Methods”: I could not catch what the authors wanted to show. Please reformat it.

Line 101-108: Not a good command of English! Please reformat it.

Line 155; Line 164-170/Line 172-177: Not a good command of English! Please reformat it.

Figure 3 & Figure 6/Figure Legend: Please reformat the legend. The original legend must be made more comprehensive. And more importantly, cut it shorter.

Figure 5: Reviewer’s Comments: I would like to simplify the figure legend as shown. Figure 5. Ouabain prevented the neurotoxicity induced by 24 hour-excitotoxic insult on the rat cortical neurons. A) Confocal image after 24 h-treatment with 100 μM homocystein+30 μM glycine (HCY); B) addition of 1 nM ouabain. Scale bar, 100 μm. Data obtained from the control and that after treated with 100 μM glutamate (Glu) or homocystein (HCY)+30 μM glycine and either 0.1 nM) or 1 nM Oua. respectively for 24 hours.

Green columns: percentages of live. Red: necrotic. Orange: apoptotic neurons. Data are expressed in Mean± SEM. *, p = 0.03; **, p = 0.002; ***, p = 0.0008, respectively, are significantly different from the control. Oua: Ouabain.

Fig. 7. Define the nomenclatures for each symbol used in this figure and interpret systematically the meaning you wante to show in

Fig. 7. Define what is “short-term” and what is ‘long-term’. I don’t agree with such a classification. The reasons are:

      Reason 7-1) Clinically, 2 h and 24 h can not be differentiated as ‘short term’ and ‘long-term’, they are all ‘acute cases”.

Reason 7-2). The expression of signals may be in a style of “sequential” or ‘serial”, some immediately response, some in delayed time region. This must be re-examined. Normally, the damaging signals that occur within the short term will be alleviated or abolished by a systemic re-accommodation of the signals appearing later and prolonged! Such a phenomenon usually occurs in the fetal or embryonic stages or cells. Or they will appear in a damping mode!

Reason 7-3: The differential drug transportation rate and up-take may trigger the downregulation of signal jMTX first and then GluA2.

Fig. 7: Normally for securing the cell viability, necrosis is harder to occur so soon, this must be explained in your text!

The dose dependent cell viability against the Glu and HCY are seen quite different from each other. Why Qua at 1 nM with HCY suppressed the viability, while Glu did not? Consequently, a serial dose-dependent cell viability on Qua is required to check this point.

Author Response

Here we provide the answers to each of comments.

General Comments:

  1. The written English has to be improved. Many points are not up to the English command. To name few and not limited these examples: Lines 12-15.

(Answer) We have re-checked the text to improve troublesome expressions.

  1. Please indicated in the “Abstract” certain sections (although hidden) involving the background, aim, and methods, results & discussion. and conclusions.

(Answer) While we agree that sections may improve understanding, the journal guidelines do not assume splitting the abstract into sections.

Specific Comments:

Line 93-96/Section 2 “Materials and Methods”: I could not catch what the authors wanted to show. Please reformat it.

(Answer) This fragment duplicates information provided below. We decided this is unnecessary and removed it.

Line 101-108: Not a good command of English! Please reformat it.

(Answer) The text was rewritten as follows:

The procedure of culture preparation from rat embryos was previously described [41,42]. Briefly, Wistar rats (overall 27 animals provided by the Sechenov Institute’s Animal Facility) 16 – 17 days pregnant were sacrificed by CO2 inhalation. Fetuses were removed and their cerebral cortices were isolated, enzymatically dissociated, and used to prepare primary neuronal cultures. Cells were grown in NeurobasalTM culture media supplemented with B-27 (Gibco) on glass coverslips coated with poly-D-lysine for 10 – 14 days (10 – 14 DIV) before experiments [41,42].

Line 155; Line 164-170/Line 172-177: Not a good command of English! Please reformat it.

(Answer) The text was rewritten as follows:

Cortical neurons in primary culture 10 – 14 days in vitro (DIV) were loaded with Fluo-8. For this reason neurons were incubated in the basic solution containing 2 µM Fluo-8 acetoxymethyl ester (Fluo-8 AM) at room temperature for 60 min. To remove contaminating dye cells were perfused with pure basic solution for 20 min. Then a coverslip with neurons was placed on the stage of a Leica TCS SP5 MP inverted microscope (Leica Microsystems, GmbH, Germany) and permanently perfused with the basic solution at a flow rate of 1.2 ml/min. The setup was equipped by the fast perfusion system, which allowed rapid application of various compounds. HCY (100 µM) or glutamate (100 µM) were added together with 30 µM glycine. Fluorescence was excited with 488 nm laser and detected at 510 - 560 nm range with ~2 s sampling interval (frame 512 x 512 px).

Cortical neurons were incubated with in the basic solution containing 5 µM Rhodamine 123 for 30 min. The dye fluorescence was detected using a Leica TCS SP5 MP inverted microscope (Leica Microsystems, Inc.). The wavelengths 488 nm and 510 - 530 nm were used for excitation and emission, respectively to monitor mitochondrial inner membrane potential (φmit). Sampling interval was set to ~1 s (frame 512 x 512 px). In experiments HCY (100 µM) or glutamate (100 µM) always were co-applied with 30 µM glycine. To test the functional state of mitochondria, the oxidative phosphorylation inhibitor, 4 µM CCCP (m-chlorophenyl hydrazone, Sigma) was added at the end of experiment [43].

Figure 3 & Figure 6/Figure Legend: Please reformat the legend. The original legend must be made more comprehensive. And more importantly, cut it shorter.

(Answer) We followed this advice of the reviewer and improved the legend to make it shorter and more comprehensive.

Figure 5: Reviewer’s Comments: I would like to simplify the figure legend as shown. Figure 5. Ouabain prevented the neurotoxicity induced by 24 hour-excitotoxic insult on the rat cortical neurons. A) Confocal image after 24 h-treatment with 100 μM homocystein+30 μM glycine (HCY); B) addition of 1 nM ouabain. Scale bar, 100 μm. Data obtained from the control and that after treated with 100 μM glutamate (Glu) or homocystein (HCY)+30 μM glycine and either 0.1 nM) or 1 nM Oua. respectively for 24 hours.

Green columns: percentages of live. Red: necrotic. Orange: apoptotic neurons. Data are expressed in Mean± SEM. *, p = 0.03; **, p = 0.002; ***, p = 0.0008, respectively, are significantly different from the control. Oua: Ouabain.

(Answer) We aggree, that this representation is better. The Figure 6 legend was rewritten accordingly.

Fig. 7. Define the nomenclatures for each symbol used in this figure and interpret systematically the meaning you wante to show in

(Answer) We added the abbreviations to the figure legend.

Fig. 7. Define what is “short-term” and what is ‘long-term’. I don’t agree with such a classification. The reasons are:

Reason 7-1) Clinically, 2 h and 24 h can not be differentiated as ‘short term’ and ‘long-term’, they are all ‘acute cases”.

Reason 7-2). The expression of signals may be in a style of “sequential” or ‘serial”, some immediately response, some in delayed time region. This must be re-examined. Normally, the damaging signals that occur within the short term will be alleviated or abolished by a systemic re-accommodation of the signals appearing later and prolonged! Such a phenomenon usually occurs in the fetal or embryonic stages or cells. Or they will appear in a damping mode!

Reason 7-3: The differential drug transportation rate and up-take may trigger the downregulation of signal jMTX first and then GluA2.

(Answer) The terms "short-term" and "long-term" in the legend were replaced with the specific periods 4 h and 24 h. These terms were introduced in the text to illustrate the difference in the protocols of excitotoxic experiments rather than the general meanings of acute or chronic experiments. We added the following sentence in “Methods” :

Line 115-116, For simplicity in further description we use name “short” and “long” to distinguish between 4 h and 24 h excitotoxicity protocols.

Fig. 7: Normally for securing the cell viability, necrosis is harder to occur so soon, this must be explained in your text!

(Answer) We always observed some minimal necrosis in normal neuronal cultures. In acute stress produced by saturating concentrations of glutamate receptor agonists we often observe rapid swelling and bursting in some neurons, which became obvious in 4 h experiments. We assume these cells to contribute to necrosis increase in our experiments. After 24 h the effect is less detectable, since the traces of necrotic cells disappear.

The dose dependent cell viability against the Glu and HCY are seen quite different from each other. Why Qua at 1 nM with HCY suppressed the viability, while Glu did not? Consequently, a serial dose-dependent cell viability on Qua is required to check this point.

(Answer) We added the reference to our previous study [31] of dose-dependence of ouabain effects in primary culture of cortical neurons (Lines 109-110). Ouabain at 0.1-1 nM provided the most pronounced neuroprotection, therefore these concentrations were used in the current study. We also previously studied the dose dependence of HCY neurotoxitity [10]. Here we use 100 µM HCY as a clinically relevant concentration observed in blood plasma in severe hyperhomocysteinemia. In spite of 500 µM HCY produced stronger neurotoxicity [10] this concentration of endogenous HCY is not physiologically probable.

Reviewer 2 Report

Main points:

Results and Figures

Figure 2: the β-actin immunoreactive bands for each condition miss. Please report.

Figure 3 and 4: Please specify the ordinate in the graph.

Figure 6: The bars related to the cell viability in the presence of glutamate and homocysteine alone (4h and 24h) and in the presence of the inhibitors of different protein kinases could help the readers.

Research design

The study is well designed but lacks some fundamental experiments:

- The effects of ouabain alone at 24h on the observed parameters are not reported.

- The Authors show the neuroprotective effect of ouabain against neurotoxicity induced by 24h homocysteine insults in rat cortical neurons with confocal images. The confocal images for the 24h glutamate exposure and the protective effect of ouabain need to be reported in Figure 5.

- In Figure 6, the Authors show the effects of inhibitors of different protein kinases on cell viability: their effects could be shown also for glutamate and homocysteine alone, in the absence of ouabain, at 4h and 24h chemical exposure.

Minor points:

Method

2.4. Western Blot Analysis: I did not find the amounts of the loaded samples on the gel in the text or the legend. Please report.

Statistical Analysis

The Authors performed experiments on n single coverslip but they did not specify if the cells derived from the same primary culture or different cultures; please specify. In the discussion they suggested that at short-term homocysteine was less potent than glutamate: a statistical analysis could be performed to compare glutamate vs homocysteine at 4h and at 24h, too.

Figure

Figures 1 and 5: A positive marker for neurons is not shown. I suggest reporting the confocal images at higher magnification with the neurons labelled with a specific marker (e.g. β-tubulin III or MAP) and the two dyes if the staining is compatible with the cell viability assay. In this way, the figures become more self – explained.

Research design

The Authors suggested a role for GluN2A in the ouabain neuroprotection activity. Experiments with a specific receptor antagonist could be performed, as well as a WB analysis to assess the level of GluA2 AMPA receptor subunit on the cell membranes after long-exposure to glutamate and homocysteine, as suggested by the same Authors.

Discussion

I was surprised that the effect of glutamate 100µM at 24h on cell vitality was not different from what observed at 4h. The Authors could suggest an explanation in the discussion.

English language and style 

Please, review the spelling (signalling or signaling? Two Ls in British English, one in American English; line 48 IC50; line 150, anti-rabbit; line 302 (c) instead (C); line 387 sort-term….). Pay attention to superscript and subscript.

Line 40…. large HCY concentrations ….

In the legend of Figure 3 and 4 … “Similar to that shown in panel”… Please rephrase the sentence…

Author Response

Here we provide the answers to each of comments.

Main points:

Results and Figures

Figure 2: the β-actin immunoreactive bands for each condition miss. Please report.

(Answer) We have added the β-actin bands to the figure.

Figure 3 and 4: Please specify the ordinate in the graph.

(Answer) We have added labels to ordinates.

Figure 6: The bars related to the cell viability in the presence of glutamate and homocysteine alone (4h and 24h) and in the presence of the inhibitors of different protein kinases could help the readers.

(Answer) We added the bands showing application of agonists alone.

Research design

The study is well designed but lacks some fundamental experiments:

- The effects of ouabain alone at 24h on the observed parameters are not reported.

(Answer) The Table 2 was added to the text, which provides the required data.

- The Authors show the neuroprotective effect of ouabain against neurotoxicity induced by 24h homocysteine insults in rat cortical neurons with confocal images. The confocal images for the 24h glutamate exposure and the protective effect of ouabain need to be reported in Figure 5.

(Answer) The corresponding confocal images were added to Figure 5.

- In Figure 6, the Authors show the effects of inhibitors of different protein kinases on cell viability: their effects could be shown also for glutamate and homocysteine alone, in the absence of ouabain, at 4h and 24h chemical exposure.

(Answer) We added the bands corresponding to application of agonists alone.

 Minor points:

Method

2.4. Western Blot Analysis: I did not find the amounts of the loaded samples on the gel in the text or the legend. Please report.

(Answer) It was 10 µg of protein. Methods at line 136 were corrected.

Statistical Analysis

The Authors performed experiments on n single coverslip but they did not specify if the cells derived from the same primary culture or different cultures; please specify. In the discussion they suggested that at short-term homocysteine was less potent than glutamate: a statistical analysis could be performed to compare glutamate vs homocysteine at 4h and at 24h, too.

(Answer) Each group contained coverslips from at least 3 different cultures. This statement was added at line 174. The observation that homocysteine was less potent than glutamate was confirmed with statistical comparison (Lines 391-393).

Figure

Figures 1 and 5: A positive marker for neurons is not shown. I suggest reporting the confocal images at higher magnification with the neurons labelled with a specific marker (e.g. β-tubulin III or MAP) and the two dyes if the staining is compatible with the cell viability assay. In this way, the figures become more self – explained.

(Answer) Unfortunately immunocytochemical detection of proteins requires fixed tissue, which is incompatible with the viability assay. However in cell cultures on glass coverslips the neuronal bodies are bulging above the surface, while glial cells are very flat. This allows placing the confocal plane to cross only neuronal bodies, which was utilized in the current study.

Research design

The Authors suggested a role for GluN2A in the ouabain neuroprotection activity. Experiments with a specific receptor antagonist could be performed, as well as a WB analysis to assess the level of GluA2 AMPA receptor subunit on the cell membranes after long-exposure to glutamate and homocysteine, as suggested by the same Authors.

(Answer) NMDA receptor inhibition with the specific antagonist was already demonstrated to rescue neurons from HCY induced stress in our previous study [10]. In discussion we cited the study [60] which show, that increased GluA2 Ca2+ permeability induced by HCY additionally enhances neuronal Ca2+ overload. We do not expect changes of GluA2 AMPA receptor expression to be the central mechanism for sustained Ca2+ elevation during long-term treatment with HCY. We decided to remove mentioning GluA2 from the Figure 7.

Discussion

I was surprised that the effect of glutamate 100µM at 24h on cell vitality was not different from what observed at 4h. The Authors could suggest an explanation in the discussion.

(Answer) This may occur because of the rundown of the neuronal population during 24 h caused by necrosis and fast developing apoptosis.

English language and style 

Please, review the spelling (signalling or signaling? Two Ls in British English, one in American English; line 48 IC50; line 150, anti-rabbit; line 302 (c) instead (C); line 387 sort-term….). Pay attention to superscript and subscript.

(Answer) The typos were corrected.

Line 40…. large HCY concentrations ….

(Answer) We meant excessive HCY concentrations. The text was corrected.

In the legend of Figure 3 and 4 … “Similar to that shown in panel”… Please rephrase the sentence…

(Answer) Figure legends were rephrased.

Reviewer 3 Report

To compare neurotoxicity caused by glutamate and HCY, two endogenous aminoacids, different molecular targets for including all ionotropic and metabotropic glutamate receptors as well as glutamate transporters should be taken into account.

In this study ouabain effects were characterized by evaluating intracellular Ca2+ signaling, mitochondrial inner membrane voltage, and the cell viability in primary cultures of rat cortical neurons in glutamate and HCY neurotoxic insults. In addition, apoptosis-related protein expression and the involvement of some kinases in ouabain mediated effects were evaluated.

In function of receptor distribution the HCY-induced neurotoxicity differs from the glutamate one in many aspects, most likely, including intracellular signaling cascades as well.

This study could be useful to characterize excitotoxic events suggesting existence of different appropriate pharmacological treatment for hyperhomocysteinemia and glutamate toxic effects.

  • It is well known that different cysteine metabolites (in particular homocysteic acid and cysteic acid)  exert toxic effects, these effects were no mentioned. Further experiments or sentences in discussion should be added.
  • Experiments to evaluate the receptor-mediated effects should be confirmed by observations with receptor-antagonists. No experiments in this field are present.
  • The mentioned “Kainite receptor” is no correct.

Author Response

We sincerely thank the reviewers for constructive criticisms and valuable comments, which were of great help in revising the manuscript.

Here we provide the answers to each of comments.

Comments and Suggestions for Authors

To compare neurotoxicity caused by glutamate and HCY, two endogenous aminoacids, different molecular targets for including all ionotropic and metabotropic glutamate receptors as well as glutamate transporters should be taken into account.

In this study ouabain effects were characterized by evaluating intracellular Ca2+ signaling, mitochondrial inner membrane voltage, and the cell viability in primary cultures of rat cortical neurons in glutamate and HCY neurotoxic insults. In addition, apoptosis-related protein expression and the involvement of some kinases in ouabain mediated effects were evaluated.

In function of receptor distribution the HCY-induced neurotoxicity differs from the glutamate one in many aspects, most likely, including intracellular signaling cascades as well.

This study could be useful to characterize excitotoxic events suggesting existence of different appropriate pharmacological treatment for hyperhomocysteinemia and glutamate toxic effects.

It is well known that different cysteine metabolites (in particular homocysteic acid and cysteic acid) exert toxic effects, these effects were no mentioned. Further experiments or sentences in discussion should be added.

(Answer) In hyperhomocysteinemia other endogenous NMDA receptor agonists like homocysteic acid (HCA) or cysteic acid can accumulate, however there is no straight correlation between HCY and HCA accumulation in CSF (Quinn at al., 1997, 10.1200/jco.1997.15.8.2800). In our experiments only HCY was added to the incubation bath. The bath volume was far bigger that the cells volume and did not allow any other metabolites to accumulate during in vitro experiments. Thus we studied pure HCY effects on neurons.

Experiments to evaluate the receptor-mediated effects should be confirmed by observations with receptor-antagonists. No experiments in this field are present.

(Answer) NMDA receptor inhibition with the specific antagonist was already demonstrated to rescue neurons from HCY induced stress in our previous study [10].

The mentioned “Kainite receptor” is no correct.

(Answer) Corrected to kainic acid. Line 67.

Round 2

Reviewer 1 Report

No further comments!

Author Response

Thank You for review.

Reviewer 2 Report

Minor points:

Line 12  :,

Line 14 in the brain. this amino acid

Line 143 10µg of each sample

Line 448 Ca2+

Legend of Figure 6: …. (B) Similar to that shown in panel (A), except the duration of excitotoxic insult is 24 h. *, **, *** - data are significantly different from the values obtained in HCY with Oua (* , p = 0.03; **, p = 0.009; ***, p = 0.0001; ANOVA with Tukey’s post-hoc test).….Please rephrase the sentence …(B) Similar to that shown in panel (A), except the duration of excitotoxic insult is 24 h.

Legend of Figure 6: The symbols were added also for the glutamate treated cells. Please define the used group for the statistical analysis.

Author Response

Thank You for review.

The typos pointed out in minor comments were corrected.

The legend for Figure 6 was rephrased as follows:

Figure 6. Effects of inhibitors of different protein kinases in ouabain induced neuroprotection.  The histograms quantitatively compare the data obtained in the presense of  100 μM glutamate (+ 30 μM glycine, Glu) or homocysteine (+ 30 μM glycine, HCY) with 1 nM ouabain (Oua),  and in combination with 0.6 μM PKA inhibitor (PKAi), 1 μM PKC inhibitor chelerythrine (Chel) or 3 μM CaMKII inhibitor (KN93) or during excitotoxic insults without Oua. Mean values ± SEM are plotted. Experimental conditions are indicated below the plot. Green columns represent percentages of live, red – of necrotic and orange – of apoptotic neurons. Data were obtained in 4 h (A) and in 24 h (B) excitoxotic insults . *, **, *** - data are significantly different from the values obtained in HCY with Oua (* , p = 0.03; **, p = 0.009; ***, p = 0.0001; ANOVA with Tukey’s post-hoc test). The number of experiments (n) for each group is indicated above the columns.

Reviewer 3 Report

I think revised form is according to referee suggestions. Now the manuscript

may be considered for publication.

Author Response

Thank You for review.